# Effects of Diffusion Coefficients and Permanent Charge on Reversal Potentials in Ionic Channels

**DOI:** 10.3390/e22030325

**Published:** 2020-03-12

**Authors:** Hamid Mofidi, Bob Eisenberg, Weishi Liu

**Affiliations:** 1Department of Mathematics, University of Kansas, Lawrence, KS 66045, USA; h.mofidi@ku.edu; 2Department of Physiology and Biophysics, Rush University, Chicago, IL 60612, USA; beisenbe@rush.edu

**Keywords:** reversal potential, effects of diffusion coefficients, permanent charge

## Abstract

In this work, the dependence of reversal potentials and zero-current fluxes on diffusion coefficients are examined for ionic flows through membrane channels. The study is conducted for the setup of a simple structure defined by the profile of permanent charges with two mobile ion species, one positively charged (cation) and one negatively charged (anion). Numerical observations are obtained from analytical results established using geometric singular perturbation analysis of classical Poisson–Nernst–Planck models. For 1:1 ionic mixtures with arbitrary diffusion constants, Mofidi and Liu (arXiv:1909.01192) conducted a rigorous mathematical analysis and derived an equation for reversal potentials. We summarize and extend these results with numerical observations for biological relevant situations. The numerical investigations on profiles of the electrochemical potentials, ion concentrations, and electrical potential across ion channels are also presented for the zero-current case. Moreover, the dependence of current and fluxes on voltages and permanent charges is investigated. In the opinion of the authors, many results in the paper are not intuitive, and it is difficult, if not impossible, to reveal all cases without investigations of this type.

## 1. Introduction

Ion channels are proteins found in cell membranes that create openings in the membrane to allow cells to communicate with each other and with the outside to transform signals and to conduct tasks together [1,2]. They have an aqueous pore that becomes accessible to ions after a change in the protein structure that makes ion channels open. Ion channels permit the selective passage of charged ions formed from dissolved salts, including sodium, potassium, calcium, and chloride ions that carry electrical current in and out of the cell.

To unravel how ion channels operate, one needs to understand the physical structure of ion channels, which is defined by the channel shape and the spatial distribution of permanent and polarization charge. The shape of a typical ion channel is often approximated as a cylindrical-like domain with a non-uniform cross-sectional area. Within a large class of ion channels, amino acid side chains are distributed mainly over a “short” and “narrow” portion of the channel, with acidic side chains contributing permanent negative charges and basic side chains contributing permanent positive charges, analogous to the doping of semiconductor devices, e.g., bipolar PNP and NPN transistors.

The spatial distribution of side chains in a specific channel defines the permanent charge of the channel. The spatial distribution of permanent charge forms (most of) the electrical structure of the channel protein. The spatial distribution of mass forms the structure studied so successfully by molecular and structural biologists. Ions that move through channels are often only an Angstrom or so away from the permanent charges residing on acid and base side chains. In addition, electrical forces are in general much stronger than entropic forces. Thus, in most cases, the electrical structure is more important in determining how ions go through a channel than the mass structure. Sometimes, the dielectric properties (“polarization”) of the protein contribute a charge that is significant. Then, the spatial distribution of dielectric properties becomes an important part of the electrical structure.

The most basic function of ion channels is to regulate the permeability of membranes for a given species of ions and to select the types of ions and to facilitate and modulate the diffusion of ions across cell membranes. At present, these permeation and selectivity properties of ion channels are usually determined from the current–voltage (I–V) relations measured experimentally [2,3]. Individual fluxes carry more information than the current, but it is expensive and challenging to measure them [4,5]. Indeed, the measurement of unidirectional fluxes by isotopic tracers allowed the early definition of channels and transporters and is a central subject in the history of membrane transport, as described in textbooks—for example, [6,7,8,9]. The precise definition and use of unidirectional fluxes are dealt with at length in the paper [5]. The I–V relation defines the function of the channel structure, namely the ionic transport through the channel. That transport depends on driving forces expressed mathematically as boundary conditions. The multi-scale feature of the problem with multiple physical parameters allows the system to have great flexibility and to exhibit vibrant phenomena/behaviors—a great advantage of “natural devices” [10]. On the other hand, the same multi-scale feature with multiple physical parameters presents an extremely challenging task for anyone to extract meaningful information from experimental data, also given the fact that the internal dynamics cannot be measured with present techniques. The general inverse problem is challenging, although specific inverse problems have been successfully solved with surprisingly little difficulty using standard methods and software packages [11].

To understand the importance of the relation of current and permanent charges, that is, the I–Q relation, we point out that the role of permanent charges in ionic channels is similar to the role of doping profiles in semiconductor devices. Semiconductor devices are similar to ionic channels in the way that they both use atomic-scale structures to control macroscopic flows from one reservoir to another. Ions move a lot like quasi-particles move in semiconductors. In a crude sense, holes and electrons are the cations and anions of semiconductors. Semiconductor technology depends on the control of migration and diffusion of quasi-particles of charge in transistors and integrated circuits. Doping is the process of adding impurities into intrinsic semiconductors to modulate its electrical, optical, and structural properties [12,13]. In a crude sense, doping provides the charges that acid and base side chains provide in a protein channel.

Ion channels are almost always passive and do not require a source of chemical energy (e.g., ATP hydrolysis) in order to operate. Instead, they allow ions to flow passively driven by a combination of the transmembrane electrical potential and the ion concentration gradient across the membrane. For other fixed physical quantities, the total current I=I(V,Q) depends on the transmembrane potential V and the permanent charge Q. For fixed Q, *a reversal potential*$V=V_{rev}\left(Q\right)$ is a transmembrane potential that produces zero current $I(V_{rev}\left(Q\right),Q)=0$. Similarly, for fixed transmembrane potential V, *a reversal permanent charge*$Q=Q_{rev}\left(V\right)$ is a permanent charge that produces zero current $I(V,Q_{rev}\left(V\right))=0$.

The *Goldman–Hodgkin–Katz (GHK) equation* for reversal potentials involving multiple ion species [14,15] is used to determine the reversal potential across ion channels. The GHK equation is an extension of the Nernst equation—the latter is for one ion species. The classical derivations were based on the incorrect assumption that the electric potential Φ(X) is linear in *X*—the coordinate along the length of the channel. This assumption is particularly unfortunate because it is the change in the shape of the electrical potential Φ(X) that is responsible for so many of the fascinating behaviors of transistors or ionic systems [16,17,18,19,20,21]. There was no substitute for GHK equations until authors of [22,23] recently offered equations derived from self-consistent Poisson–Nernst–Planck (PNP) systems, to the best of our knowledge.

In this work, focusing on basic understanding of possible effects of unequal diffusion coefficients and, as a starting point, we will use the classical PNP model with a piecewise constant permanent charge and a cylinder-like channel with variable cross-sectional area. The classical PNP model treats ions as point charges. Among many limitations, gating and selectivity cannot be captured by the simple classical PNP model. However, the basic finding on reversal potentials and their dependence on permanent charges and on ratios of diffusion constants seems important and some are non-intuitive and deserving of further investigation. In the future, more structural detail and more correlations between ions should be taken into considerations in PNP models such as those including various potentials for ion-to-ion interaction accounting for ion sizes effects and voids [24,25,26,27,28,29,30,31,32].

There have been great achievements in analyzing the PNP models for ionic flows through ion channels [5,28,33,34,35,36], etc. Although mathematical analysis plays a powerful and unique role to explain mechanisms of observed biological phenomena and to discover new phenomena, numerical simulations are needed to fit actual experimental data and study cases where analytical solutions do not exist. Furthermore, numerical observations may give clues for more theoretical investigations. Indeed, numerical and analytical studies are linked; any progress in one catalyzes work in the other.

This paper is a mathematical study on some aspects of ionic flows via the PNP models. It uses established mathematical methods and analytical results [23,33] that are derived without further assumption from their underlying physical models. The numerical results, throughout the paper, are gained from the algebraic systems (Equation 15), (Equation 22), (Equation 23) and (Equation 27), obtained from reduced matching systems of analytical results in [23,33]. The nonlinear algebraic systems are then solved by the MATLAB^®^ (Version 9.5) function *fsolve* that uses the trust–region dogleg algorithm. The trust–region algorithm is a subspace trust–region method and is based on the interior-reflective Newton method described in [37]. Our numerical results indicate that current–voltage and current-permanent charge and even zero-current relations depend on a rich interplay of boundary conditions and the channel geometry arising from the mathematical properties analyzed in [23,33,34,38]. Although the work here is presented in the context of biological ion channels, it is clear that the results apply to the artificial channels that are now being studied for their engineering applications.

The highlights of our studies in this paper as well as in [23,33,34,38] applied to the setup of this paper include:(i)a mathematically derived system for the zero-current condition (see System (Equation 15)) that can be used to determine the reversal potential in terms of other parameters (see Display (Equation 22));(ii)an examination on how the reversal potential depends on permanent charge: its sign and its monotonicity in permanent charge (see Section 2.2); and a comparison between this reversal potential and that from GHK in the special setting (see Section 2.3);(iii)a characterization of monotonic dependencies of the reversal potential on the ratio of diffusion coefficients in terms of different conditions on the boundary concentrations (see Section 2.2), as well as effects of un-equal diffusion coefficients on signs of zero-current flux and its dependence on permanent charge (see Section 2.1);(iv)numerical spatial profiles under the zero current condition of the concentrations and electric potential, and hence the profiles of the electrochemical potentials for several choices of permanent charges that reveal special features of permanent charge effects (see Section 2.4, particularly, Remark 3);(v)numerical and analytical studies of I–V and I–Q relations, and zero-voltage current and its rich dependence on permanent charge (see Section 3.3).

Furthermore, there are several qualitatively important but non-intuitive results discussed in this work. These qualitative results may be helpful in guiding experimentation and some might not be apparent in intuitive thinking about ion channel behavior. Here are some examples:a.The zero-current flux *J* has the same sign as that of l−r (see Section 2.1).b.The magnitude of the ratio between of the two diffusion coefficients affects the monotonicity of the zero-current flux *J* in *Q* (see Section 2.1).c.I–Q curves are not monotonic in general (see Section 3.2).d.Rich phenomena of interplay between boundary conditions and diffusion coefficients in terms of monotonicity of zero-voltage current on permanent charge (Section 3.3).

To this end, we would like to emphasize that applying the geometric analysis allows us to identify and formulate quantities and properties that are crucial to biology, while also providing quantitative and qualitative understanding and predictions.

This paper is organized as follows. The classical PNP model for ionic flows is recalled in Section 1.1 to prepare the stage for investigations in later sections. In Section 2, we study zero current problems to investigate the corresponding fluxes and reversal potentials $V_{rev}$. In particular, we compare a special case of the reversal potential with the GHK equation. Some other numerical observations are also provided to study profiles of relevant physical quantities in Section 2.4. In Section 3, we first recall the analytical results in [33] when diffusion constants are also involved. Then, numerical observations are provided to examine behaviors of current, voltage, and permanent charge with respect to each other in some general cases. Some concluding remarks are provided in Section 4.

### 1.1. Poisson–Nernst–Planck Models for Ionic Flows

The PNP system of equations has been analyzed mathematically to some extent, but the equations have been simulated and computed to a much larger extent [39,40,41,42,43]. One can see from these simulations that macroscopic reservoirs must be included in the mathematical formulation to describe the actual behavior of channels [24,44]. For an ionic mixture of *n* ion species, the PNP type model is, for k=1,2,…,n,
(1)$$\begin{array}{c} \begin{array}{cc} \mathrm{Poisson}:\; & \nabla \cdot \left(\varepsilon_{r}\left(\vec{X}\right)\varepsilon_{0}\nabla \Phi \right)=-e_{0}\left(\sum_{s=1}^{n}z_{s}C_{s}+Q\left(\vec{X}\right)\right),\\ \mathrm{Nernst}-\mathrm{Planck}:\; & \partial_{t}C_{k}+\nabla \cdot {\vec{J}}_{k}=0,\;-{\vec{J}}_{k}=\frac{1}{k_{B}T}D_{k}\left(\vec{X}\right)C_{k}\nabla \mu_{k}, \end{array} \end{array}$$
where $\vec{X}\in \Omega$ with Ω being a three-dimensional cylindrical-like domain representing the channel of length $\hat{L}\;\left(\mathrm{nm}=\hat{L}\times 10^{-9}m\right)$, $Q(\vec{X})$ is the permanent charge density of the channel (with unit $1M=1\mathrm{Molar}=1\mathrm{mol}/L=10^{3}\mathrm{mol}/m^{3}$), $\varepsilon_{r}\left(\vec{X}\right)$ is the relative dielectric coefficient (with unit 1), $\varepsilon_{0}\approx 8.854\times 10^{-12}\;{Fm}^{-1}$ is the vacuum permittivity, $e_{0}\approx 1.602\times 10^{-19}C$ (coulomb) is the elementary charge, $k_{B}\approx 1.381\times 10^{-23}{\mathrm{JK}}^{-1}$ is the Boltzmann constant, *T* is the absolute temperature (T≈273.16K= kelvin, for water); Φ is the electric potential (with the unit $V=\mathrm{Volt}={\mathrm{JC}}^{-1}$), and, for the *k*-th ion species, $C_{k}$ is the concentration (with unit M), $z_{k}$ is the valence (the number of charges per particle with unit 1), and $\mu_{k}$ is the electrochemical potential (with unit J=CV) depending on electrical potential Φ and concentrations $C_{k}$. The flux density ${\vec{J}}_{k}\left(\vec{X}\right)$ (with unit $\mathrm{mol}m^{-2}s^{-1}$) is the number of particles across each cross-section in per unit time, $D_{k}\left(\vec{X}\right)$ is the diffusion coefficient (with unit $m^{2}/s$), and *n* is the number of distinct types of ion species (with unit 1).

Ion channels have narrow cross-sections relative to their lengths. Therefore, three-dimensional PNP type models can be reduced to quasi-one-dimensional models. The authors of [45] first offered a reduced form, and, for a particular case, the reduction is precisely verified by the mathematical analysis of [46]. The quasi-one-dimensional steady-state PNP type is, for k=1,2,…,n,
(2)$$\begin{array}{c} \begin{array}{cc} \frac{1}{A(X)}\frac{d}{dX}\left(\varepsilon_{r}\left(X\right)\varepsilon_{0}A\left(X\right)\frac{d\Phi }{dX}\right)= & -e_{0}\left(\sum_{s=1}^{n}z_{s}C_{s}+Q\left(X\right)\right),\\ \frac{dJ_{k}}{dX}=0,\;-J_{k}= & \frac{1}{k_{B}T}D_{k}\left(X\right)A\left(X\right)C_{k}\frac{d\mu_{k}}{dX}, \end{array} \end{array}$$
where *X* is the coordinate along the channel, A(X) is the area of cross-section of the channel over location *X*, and $J_{k}$ (with unit $\mathrm{mol}s^{-1}$) is the total flux through the cross-section. Equipped with System (Equation 2), we impose the following boundary conditions, for k=1,2,⋯,n,
(3)$$\Phi \left(0\right)=V,\;C_{k}\left(0\right)=L_{k}>0;\;\Phi \left(\hat{L}\right)=0,\;C_{k}\left(\hat{L}\right)=R_{k}>0.$$

One often uses the electroneutrality conditions on the boundary concentrations because the solutions are made from electroneutral solid salts,
(4)$$\begin{array}{c} \sum_{s=1}^{n}z_{s}L_{s}=\sum_{s=1}^{n}z_{s}R_{s}=0. \end{array}$$

The electrochemical potential $\mu_{k}\left(X\right)$ for the *k*-th ion species consists of the ideal component $\mu_{k}^{id}\left(X\right)$ and the excess component $\mu_{k}^{ex}\left(X\right)$, i.e., $\mu_{k}\left(X\right)=\mu_{k}^{id}\left(X\right)+\mu_{k}^{ex}\left(X\right)$. The excess electrochemical potential $\mu_{k}^{ex}\left(X\right)$ accounts for the finite size effect of ions. It is needed whenever concentrations exceed, say 50 mM, as they almost always do in technological and biological situations and often reach concentrations 1M or more. The classical PNP model only deals with the ideal component $\mu_{k}^{id}\left(X\right)$, which reflects the collision between ions and water molecules and ignores the size of ions; that is,
(5)$$\mu_{k}\left(X\right)=\mu_{k}^{id}\left(X\right)=z_{k}e_{0}\Phi \left(X\right)+k_{B}T\ln\frac{C_{k}\left(X\right)}{C_{0}},$$
where $C_{0}$ is a characteristic concentration of the problems, and one may consider
(6)$$C_{0}=\max_{1\leq k\leq n}\{L_{k},R_{k},\underset{X\in [0,\hat{L}]}{\sup}\left|Q\left(X\right)\right|\}.$$

For given V, Q(X), $L_{k}$’s and $R_{k}$’s, if $\left(\Phi \left(X\right),C_{k}\left(X\right),J_{k}\right)$ is a solution of the boundary value problem (BVP) (Equation 2) and (Equation 3), then the electric current I is
(7)$$\begin{array}{c} I=e_{0}\sum_{s=1}^{n}z_{s}J_{s}. \end{array}$$

For an analysis of the BVP (Equation 2) and (Equation 3), we work on a dimensionless form. Set
$$D_{0}=\max_{1\leq k\leq n}\left\{\underset{X\in [0,\hat{L}]}{\sup}D_{k}\left(X\right)\right\}\;\mathrm{and}\;{\bar{\varepsilon }}_{r}=\underset{X\in [0,\hat{L}]}{\sup}\varepsilon_{r}\left(X\right).$$

Let
(8)$$\begin{array}{cc} \; & \varepsilon^{2}=\frac{{\bar{\varepsilon }}_{r}\varepsilon_{0}k_{B}T}{e_{0}^{2}{\hat{L}}^{2}C_{0}},\;{\hat{\varepsilon }}_{r}\left(x\right)=\frac{\varepsilon_{r}\left(X\right)}{{\bar{\varepsilon }}_{r}},\;x=\frac{X}{\hat{L}},\;h\left(x\right)=\frac{A(X)}{{\hat{L}}^{2}},\;D_{k}\left(x\right)=\frac{D_{k}\left(X\right)}{D_{0}},\\ \; & Q\left(x\right)=\frac{Q(X)}{C_{0}},\;\phi \left(x\right)=\frac{e_{0}}{k_{B}T}\Phi \left(X\right),\;c_{k}\left(x\right)=\frac{C_{k}\left(X\right)}{C_{0}},\;{\hat{\mu }}_{k}=\frac{1}{k_{B}T}\mu_{k},\;J_{k}=\frac{J_{k}}{\hat{L}C_{0}D_{0}}. \end{array}$$

In terms of the new variables, the BVP (Equation 2) and (Equation 3) becomes, for k=1,2,⋯,n,
(9)$$\begin{array}{c} \begin{array}{cc} \frac{\varepsilon^{2}}{h(x)}\frac{d}{dx}\left({\hat{\varepsilon }}_{r}\left(x\right)h\left(x\right)\frac{d}{dx}\phi \right)= & -\sum_{s=1}^{n}z_{s}c_{s}-Q\left(x\right),\\ \frac{dJ_{k}}{dx}=0,\;-J_{k}= & h\left(x\right)D_{k}\left(x\right)c_{k}\frac{d}{dx}{\hat{\mu }}_{k}, \end{array} \end{array}$$
with the boundary conditions
(10)$$\phi \left(0\right)=V=\frac{e_{0}}{k_{B}T}V,\;c_{k}\left(0\right)=l_{k}=\frac{L_{k}}{C_{0}};\;\phi \left(1\right)=0,\;c_{k}\left(1\right)=r_{k}=\frac{R_{k}}{C_{0}}.$$

**Remark** **1.**
*The actual dimensional forms of quantities have been used for all figures throughout the paper, that is,*
(11)$$\begin{array}{cc} C_{k}= & C_{0}c_{k}\;\left(M\right),\;Q=C_{0}Q\;\left(M\right),\;\mu_{k}=\left(e_{0}\Phi +k_{B}T\ln\left(C_{k}/C_{0}\right)\right)\;\left(J\right),\\ \Phi = & \frac{k_{B}T}{e_{0}}\phi \;\left(V\right),\;J_{k}=\hat{L}C_{0}D_{0}J_{k}\;\left(mol/s\right),\;I=\hat{L}C_{0}D_{0}e_{0}I\;\left(A\right), \end{array}$$
*and we take $C_{0}=10\;M$, $\hat{L}=2.5\;nm$ and $D_{0}=2.032\times 10^{-9}\;m^{2}/s,$ and, for diffusion constants [31],*
(12)$$\begin{array}{cc} D_{k}= & 1.334\times 10^{-9}\;m^{2}/s\;for\;{Na}^{+},\;or\\ D_{k}= & 2.032\times 10^{-9}\;m^{2}/s\;for\;{Cl}^{-},\;\;or\\ D_{k}= & 0.792\times 10^{-9}\;m^{2}/s\;for\;{Ca}^{2+}. \end{array}$$


### 1.2. Setup of the Problem

We now designate the case we will study in this paper. We will investigate a simple setup, the classical PNP model (Equation 9) with ideal electrochemical potential (Equation 5), and the boundary conditions (Equation 10). More precisely, we assume

(A0)The ionic mixture consists of two ion species with valences $z_{1}=-z_{2}=1$;(A1)$D_{k}\left(x\right)=D_{k}$ for k=1,2 is a constant and $\hat{\varepsilon }\left(x\right)=1$;(A2)Electroneutrality boundary conditions (Equation 4) hold;(A3)The permanent charge *Q* is piecewise constant with one nonzero region; that is, for a partition 0<a<b<1 of [0,1],
(13)$$\begin{array}{l} Q\left(x\right)=\left\{\begin{array}{ll} Q_{1}=Q_{3}=0, & x\in (0,a)\cup (b,1),\\ Q_{2}=2Q_{0}, & x\in (a,b), \end{array}\right. \end{array}$$
where $Q_{0}$ is a constant.

We assume that ε>0 in System (Equation 14) is small. The assumption is reasonable since, if $\hat{L}=2.5\;\mathrm{nm}=2.5\times 10^{-9}\;m$ and $C_{0}=10\;M$, then $\varepsilon \approx 10^{-3}$ [47]. The assumption that ε is small enables one to treat System (Equation 14) of the dimensionless problem as a singularly perturbed problem that can be analyzed by the theory of geometrical singular perturbations (GSP). GSP uses the modern invariant manifold theory from nonlinear dynamical system theory to study the entire structure, i.e., the phase space portrait of the dynamical system, and is not to be confused with the classical singular perturbation theory that uses, for example, matched asymptotic expansions.

We rewrite the classical PNP system (Equation 9) into a standard form of singularly perturbed systems and turn the boundary value problem to a connecting problem. We refer the readers to the papers [33] and [36] (with insignificantly altered notations) for details. Denote the derivative with respect to *x* by overdot and introduce $u=\varepsilon \dot{\phi }$. System (Equation 9) becomes, for k=1,2,
(14)$$\begin{array}{c} \begin{array}{cc} \varepsilon \dot{\phi }= & u,\;\varepsilon \dot{u}=-\sum_{s=1}^{2}z_{s}c_{s}-Q\left(x\right)-\varepsilon \frac{h_{x}\left(x\right)}{h(x)}u,\\ \varepsilon {\dot{c}}_{k}= & -z_{k}c_{k}u-\varepsilon \frac{J_{k}}{D_{k}h\left(x\right)},\;{\dot{J}}_{k}=0. \end{array} \end{array}$$

System (Equation 14) will be treated as a dynamical system with the phase space $R^{7}$ and the independent variable *x* is viewed as time for the dynamical system.

A GSP framework for analyzing BVP of the classical PNP systems was developed first in [33,35] for ionic mixtures with two types of ion species. The model of ion channel properties involves coupled nonlinear differential equations. Until accomplished, it was not apparent that any analytical results could be found, let alone the powerful ones provided by geometrical singular perturbation. This GSP framework was extended to an arbitrary number of types of ion species successfully only when *two special mathematical structures* of the PNP system were revealed [36]. One special structure is *a complete set of integrals (or conserved quantities)* for the ε=0 limit fast (or inner) system that allows a detailed analysis of a singular layer component of the full problem. It should pointed out that most of the integrals are NOT conserved for the physical problem since, no matter how small it is, ε is NOT zero. The GSP allows one to make conclusion about the BVP for ε>0 small from information of ε=0 limit systems. The other special structure is that *a state-dependent scaling* of the independent variable turns the nonlinear limit slow (or outer) system to a linear system with constant coefficients. The coefficients do depend on unknown fluxes to be determined as a part of the whole problem, and this is the mathematical reason for the rich dynamics of the problem. As a consequence of the framework, the existence, multiplicity, and spatial profiles of *the singular orbits*—zeroth order in ε approximations of the BVP—are reduced to a system of nonlinear algebraic equations that involves all relevant quantities altogether. This system of nonlinear **algebraic** equations defines the physical framework of the problem precisely. The system shows explicitly what has been guessed implicitly “everything interacts with everything else” and, in the cases analyzed in this paper, the system shows quantitatively how those interactions occur. This geometric framework with its extensions to include some of the effects of ion size [28,29,32] has produced a number of results that are central to ion channel properties [5,23,30,34,38,48]; for example, it was shown in [34] that *a positive permanent charge may enhance anion flux as well as cation flux*; and, *in order to optimize effects of the permanent charge, the channel should have a short and narrow neck within which the permanent charge is confined*; and, it was shown in [38] that *large permanent charge is responsible for the declining phenomenon—decreasing flux with increasing transmembrane electrochemical potential*. We refer the readers to the aforementioned papers for more details on geometric singular perturbation framework for PNP as well as concrete applications to ion channel problems.

In this paper, we will apply some results and follow the notations in [23,33] for analytical results where the quantities are all in their dimensionless forms. In addition, for simplicity, we use the letters *l*, *r* and $Q_{0}$ where $l_{1}=l_{2}=l$, $r_{1}=r_{2}=r$, $Q_{2}=2Q_{0}$.

**Remark** **2.**
*We remind the readers that the quantities $V,l,r,c_{k},Q,\phi ,{\hat{\mu }}_{k},J_{k},D_{k}$, and I are dimensionless quantities corresponding to the dimensional quantities $V,L,R,C_{k},Q,\Phi ,\mu_{k},J_{k},D_{k}$, and I, respectively, obtained from Display *(Equation 8)*. We switch from dimensional form to the dimensionless form and vice versa several times throughout the paper. Dimensionless variables are convenient for illustrating and analyzing mathematical and general physical relations. Dimensional variables are necessary for showing how evolution has exploited those general relations.*


## 2. Zero Current Problems with General Diffusion Constants

In this section, we study how boundary concentrations, electric potential, permanent charges, and diffusion constants work together to produce current reversal. Throughout this section, in order to express the effects of diffusion constants on zero-current flux and reversal potential, we study and compare the results for different cases of diffusion constants where $D_{1}=D_{2}$ and where $D_{1}\neq D_{2}$, to indicate and emphasize the differences.

Diffusion is the phenomenon through which the spatial distribution of solute particles varies as a result of their potential energy. It is a spontaneous process that acts to eliminate differences in concentration and eventually leads a given mixture to a state of uniform composition. Fick’s first law [49] describes diffusion of uncharged particles by $\partial_{t}c=D\partial_{xx}^{2}c$, where *c* is the concentration, D is the diffusion constant, and *t* is time. Frequently, the determination of diffusion constants involves measuring sets of simultaneous values of *t*, *c*, and *x*. These measured values are then applied to a solution of Fick’s law to get the diffusion constants. Many techniques are available for the determination of diffusion constants of ions (charge particles) in aqueous solutions [31,50,51,52,53], etc. When diffusion constants are equal, classical electrochemistry tells that many electrical phenomena “disappear” altogether, e.g., the “liquid junction” is zero. If the diffusion constants of potassium and chloride are equal, classical electrochemistry says that KCl acts nearly as an uncharged species. Indeed, this is the basis for the saturated KCl salt bridge used in a broad range of electrochemical experiments for many years. Therefore, the equal diffusion constants case is quite degenerate. Experimental measurements are exclusively performed under isothermal conditions to avoid deviation of D values. Nevertheless, even diffusion constants of certain ionic species may differ from one method to another, even when all other parameters are held constant. Everything becomes much more complicated mathematically when the diffusion constants are not equal, however. This complexity is what makes many biological and technological devices interesting, useful, and valuable. Some kinds of selectivity depend on the non-equality of diffusion constants as well.

Applying GSP theory to the classical PNP system (Equation 2) for two ion species with diffusion constants $D_{k},\;k=1,2$, the authors of [23] obtained an algebraic matching system with eleven equations and eleven unknowns for zero current problems and singular orbits on [0,1]. They further reduced the matching system for the case where two ion valences satisfy $z_{1}=-z_{2}$. It follows that the reduced matching system for zero current $I=J_{1}-J_{2}=0$ when $z_{1}=-z_{2}=1$ is
(15)$$\begin{array}{c} G_{1}\left(A,Q_{0},\theta \right)=V\;\mathrm{and}\;G_{2}\left(A,Q_{0},\theta \right)=0, \end{array}$$
where
(16)$$\begin{array}{ll} G_{1}\left(A,Q_{0},\theta \right)= & \theta \left(\ln\frac{S_{a}+\theta Q_{0}}{S_{b}+\theta Q_{0}}+\ln\frac{l}{r}\right)-\left(1+\theta \right)\ln\frac{A}{B}+\ln\frac{S_{a}-Q_{0}}{S_{b}-Q_{0}},\\ G_{2}\left(A,Q_{0},\theta \right)= & \theta Q_{0}\ln\frac{S_{a}+\theta Q_{0}}{S_{b}+\theta Q_{0}}-N, \end{array}$$
and, *A* is the geometric mean of concentrations at x=a, that is,
(17)$$A=\sqrt{c_{1}\left(a\right)c_{2}\left(a\right)},$$
(18)$$B=\frac{1-\beta }{\alpha }\left(l-A\right)+r,\;S_{a}=\sqrt{Q_{0}^{2}+A^{2}},\;S_{b}=\sqrt{Q_{0}^{2}+B^{2}},\;N=A-l+S_{a}-S_{b},$$
and
(19)$$\theta =\frac{D_{2}-D_{1}}{D_{2}+D_{1}},\;\alpha =\frac{H(a)}{H(1)},\;\beta =\frac{H(b)}{H(1)}\;\mathrm{where}\;H\left(x\right)=\int_{0}^{x}\frac{1}{h(s)}ds.$$

Note that, if h(x) is uniform, then H(x) is the ratio of the length with the cross-section area of the potion of the channel over [0,x]. The original of this quantity H(x) has its root in Ohm law for resistance of a uniform resistor. It turns out that the quantities α and β together with the value $Q_{0}$ are key characteristics for the shape and the permanent charge of the channel structure (see Section 4 in [34] for more detailed and concrete results about the roles of α and β on the fluxes).

To this end, we recall three relevant results from [23] on which most of our analytical and numerical studies are based.

For fixed $Q_{0}$ and θ, *A* can actually be solved from $G_{2}\left(A,Q_{0},\theta \right)=0$, where $G_{2}$ is defined in Display (Equation 16) with the properties stated in the next theorem.

**Theorem** **1**(Theorem 3.4 in [23]). *The solution $A=A(Q_{0},\theta )$ of $G_{2}\left(A,Q_{0},\theta \right)=0$ satisfies*
*(a)* A(0,θ)=(1−α)l+αr and $\lim_{Q_{0}\to \pm \infty }A\left(Q_{0},\theta \right)=l$,*(b)* if l>r, then $l>A\left(Q_{0},\theta \right)>A^{*}>B\left(Q_{0},\theta \right)>r$,*(c)* if l<r, then $l<A\left(Q_{0},\theta \right)<A^{*}<B\left(Q_{0},\theta \right)<r$,*(d)* if $\theta Q_{0}\geq 0$, then $\partial_{Q_{0}}A\left(Q_{0},\theta \right)$ has the same sign as that of $\left(l-r\right)Q_{0}$,
*where $A^{*}=\frac{(1-\beta )l+\alpha r}{1-\beta +\alpha }$.*


For fixed $Q_{0}$ and θ, the reversal potential $V_{rev}=V_{rev}\left(Q_{0},\theta \right)$ can also be determined and enjoy properties stated in the next two theorems. Recall that we denote $J_{1}=J_{2}$ by *J*.

**Theorem** **2**(Theorem 4.2 in [23]). *For the reversal potential $V_{rev}=V_{rev}\left(Q_{0},\theta \right)$, one has*
*(i)* if l>r, then J>0, and, hence, $-\frac{1}{z_{1}}\ln\frac{l}{r}<V_{rev}\left(Q_{0},\theta \right)<\frac{1}{z_{1}}\ln\frac{l}{r}$;*(ii)* if l<r, then J<0, and, hence, $\frac{1}{z_{1}}\ln\frac{l}{r}<V_{rev}\left(Q_{0},\theta \right)<-\frac{1}{z_{1}}\ln\frac{l}{r}$;*(iii)* $V_{rev}\left(0,\theta \right)=\frac{\theta }{z_{1}}\ln\frac{l}{r}$ and $\lim_{Q_{0}\to \pm \infty }V_{rev}\left(Q_{0},\theta \right)=\pm \frac{1}{z_{1}}\ln\frac{l}{r}$.

**Theorem** **3**(Theorem 4.4 in [23]). *For any given θ∈(−1,1), one has*
*(i)* if θ=0, then $V_{rev}\left(Q_{0},\theta \right)$ is increasing in $Q_{0}$ for l>r and decreasing in $Q_{0}$ for l<r;*(ii)* if θ>0, then, for $Q_{0}\geq 0$, $V_{rev}\left(Q_{0},\theta \right)$ is increasing in $Q_{0}$ for l>r and decreasing in $Q_{0}$ for l<r;*(iii)* if θ<0, then, for $Q_{0}\leq 0$, $V_{rev}\left(Q_{0},\theta \right)$ is increasing in $Q_{0}$ for l>r and decreasing in $Q_{0}$ for l<r.

In what follows, numerical simulations are conducted with the help of analysis on System (Equation 15). The combination of numerics and analysis gives a better understanding of the zero-current problems and compliments some analytical results obtained in [23]. For our numerical simulations, we choose a=1/3, b=2/3 in Display (Equation 13) and h(x)=1 for simplicity and for definiteness.

### 2.1. Zero-Current Flux $J=J_{1}=J_{2}$.

We aim to clarify the relationships of ion fluxes with permanent charge and diffusion constants when current is zero.

Recall that fluxes $J_{1}$ and $J_{2}$ are equal for this case and let *J* denote it. For any permanent charge $Q=2Q_{0}$, once a solution (A,V) of System (Equation 15) is obtained, it follows from matching equations (see Appendix in [23]) that *J* is given by
(20)$$J=-\frac{6D_{1}D_{2}\left(A-l\right)}{\left(D_{1}+D_{2}\right)}=-\frac{6D_{1}D_{2}\left(r-B\right)}{\left(D_{1}+D_{2}\right)}.$$

#### 2.1.1. Sign of Zero-Current Flux J

It was observed in [22] that the Nernst–Planck equation in Display (Equation 9) (with dimensionless quantities) gives, for k=1,2,
(21)$$\begin{array}{c} \frac{J_{k}}{D_{k}}\int_{0}^{1}\frac{1}{h\left(x\right)c_{k}\left(x\right)}dx=z_{k}V+\ln\frac{l}{r}. \end{array}$$

Therefore, the sign of flux $J_{k}$ depends only on the boundary conditions *l*, *r* and *V*. Note that Equation (Equation 21) holds for any condition, not just zero-current condition.

For zero-current problem, $V=V_{rev}$ depends on *l*, *r*, $D_{1}$, $D_{2}$, and *Q* as well, in general. Thus, the sign of zero-current flux *J* seems to depend on all quantities and to be difficult to figure out. It is not the case. A consequence of Display (Equation 20) together with Theorem 1 is that:Thezero−currentfluxJhasthesamesignasthatofl−r

The latter follows directly from Theorem 1 that, for zero-current, l−A has the same sign as that of l−r. This is consistent with observations in Figure 1 where $D_{1}=1.334\times 10^{-9}\;m^{2}/s$ is fixed, and $D_{2}$ varies from the same value to $D_{2}=2.032\times 10^{-9}\;m^{2}/s$, and to a random large value.

#### 2.1.2. Dependence of Zero-Current Flux *J* on $Q_{0}$ and $D_{k}$’s

Concerning the dependence of the zero-current flux *J* on $Q_{0}$, we have the following:(i)If $D_{1}=D_{2}$, then the zero-current flux J is an even function in $Q_{0}$, and it is monotonic for $Q_{0}>0$.In this case, θ=0 and, hence, it follows from Theorem 1 that *A* is an even function in $Q_{0}$ and is monotonic in $Q_{0}$ for $Q_{0}>0$, and thus is the zero-current flux *J* from Display (Equation 20).(ii)If $D_{1}\neq D_{2}$, then the zero-current flux J is not an even function in $Q_{0}$ and the monotonicity of the zero-current flux J in $Q_{0}$ seems to be more complicated.In this case, it can be seen that $G_{2}$ in Display (Equation 16) is not an even function in $Q_{0}$, and, hence, the zero-current flux *J* is not. We would like to point out that, it follows from [38], for fixed $\rho =D_{2}/D_{1}$, no matter how large, one always has J→0 as $Q_{0}\to \pm \infty$ that is consistent with the observations in Figure 1.(iii)Another fascinating result is that *the magnitude of $\rho =D_{2}/D_{1}$ affects the monotonicity of the zero-current flux J in $Q_{0}$.*In this case, if one fixes $D_{1}$, and let $D_{2}$ increases from small values to $D_{2}\to \infty$, (i.e., ρ→∞), then it follows from Display (Equation 20) that there is a meaningful change in the monotonicity of the zero-current flux *J*, for small values of $Q_{0}$ that is not intuitive.

Let us consider the case where L<R and $Q_{0}<0$ is small. Recall that *A* is the geometric mean of concentrations at x=a. It follows from System (Equation 15) and (Equation 16) that, as $Q_{0}$ increases,
(a)*A* increases if ρ≈1 (that is θ≈0), and consequently the zero-current flux *J* decreases;(b)*A* decreases if ρ≫1 (that is θ≫1), and, hence, the zero-current flux *J* increases.

Thus, depending on the size of ρ, the zero-current flux *J* may increase or decrease in $Q_{0}<0$, which is also consistent with the observations in Figure 1. The analysis for the case with L>R is similar.

It seems likely that the engineering, like evolution, will use these mathematical properties to control the qualitative properties of channels, technological, and biological.

### 2.2. Reversal Potential $V_{rev}$.

Experimentalists have long identified reversal potential as an essential characteristic of ion channels [54,55]. Reversal potential is the potential at which the current reverses direction, i.e., $V=\Phi \left(0\right)-\Phi \left(\hat{L}\right)$ that produces zero current I. Using dimensionless form of quantities (see Remark 2), it follows from System (Equation 15) and (Equation 16) (where there are two ion species with valences $z_{1}=-z_{2}=1$) that for general permanent charge $Q=2Q_{0}\neq 0$ with arbitrary diffusion constants [23], the variable *A* (the geometric mean of concentrations at x=a) can be solved uniquely from $G_{2}=0$ in System (Equation 15), and the reversal potential is then
(22)$$\begin{array}{c} V_{rev}=\theta \left(\ln\frac{S_{a}+\theta Q_{0}}{S_{b}+\theta Q_{0}}+\ln\frac{l}{r}\right)-\left(1+\theta \right)\ln\frac{A(Q_{0},\theta )}{B(Q_{0},\theta )}+\ln\frac{S_{a}-Q_{0}}{S_{b}-Q_{0}}, \end{array}$$
where *B*, $S_{a}$, $S_{b}$, and θ are defined in Displays (Equation 18) and (Equation 19).

#### 2.2.1. Range of Reversal potential $V_{rev}$

For fixed *l*, *r*, and for any given $Q_{0}$, it follows from Theorem 2 that there exists a unique reversal potential $V_{rev}$ such that $V_{rev}\leq \left|\ln\frac{l}{r}\right|$. As $Q_{0}\to \pm \infty$, then $V_{rev}$ gets close to the boundary values, i.e., $V_{rev}\to \pm \ln\frac{l}{r}$.

#### 2.2.2. Zero Reversal Potential

One particular case is when the reversal potential is zero. To examine under what conditions one obtains $V_{rev}=V_{rev}\left(Q_{0}\right)=0$, it follows Theorem 2 that,
(i)if $D_{1}=D_{2}$, then $V_{rev}\left(Q_{0}\right)=0$ for $Q_{0}=0$,(ii)if $D_{1}<D_{2}$, then there is a $Q_{0}<0$, such that $V_{rev}\left(Q_{0}\right)=0$,(iii)if $D_{1}>D_{2}$, then there is a $Q_{0}>0$, such that $V_{rev}\left(Q_{0}\right)=0$.

Considering the second case above, the observations in Figure 2 show that, as $\rho =D_{2}/D_{1}$ increases, magnitude of the corresponding $Q_{0}$ becomes larger. In fact, as ρ→∞, then $Q_{0}\to -\infty$.

#### 2.2.3. Reversal Potential $V_{rev}\left(Q_{0}\right)$ for $Q_{0}=0$

For $Q_{0}=0$, one has $V_{rev}\left(0\right)=\theta \ln\frac{l}{r}$ from Theorem 2. Therefore,
(i)if $D_{1}=D_{2}$, then $V_{rev}\left(0\right)=0$,(ii)if $D_{1}\neq D_{2}$, then $V_{rev}\left(0\right)$ has the same sign as that of θ(l−r).

Let us consider the case where $D_{1}<D_{2}$. In that case, $V_{rev}\left(0\right)$ has the same sign as that of l−r. This is reasonable, since, for V=0, we have $|J_{1}\left|<\right|J_{2}|$ (since all but $J_{k}/D_{k}$ are independent of $D_{k}$ in Equation (Equation 21)), and to help $|J_{1}|$ more than $|J_{2}|$ to get $J_{1}=J_{2}$ for zero current conditions, one needs to increase *V* when l>r (and decrease *V* when l<r), and that is why $V_{rev}\left(0\right)>0$ for l>r (and $V_{rev}\left(0\right)<0$ for l<r). This is consistent with observations in Figure 2 as well. The analysis for the other case with $D_{1}>D_{2}$ is similar.

#### 2.2.4. Monotonicity of $V_{rev}$ with Respect to Q

It follows from Theorem 3 that
$$For\;\theta Q>0,\;\partial_{Q}V_{rev}\;has\;the\;same\;sign\;as\;that\;of\;l-r.$$

This analytical result does not allow conclusions about the case for θQ<0, however. The observations in Figure 2 and Figure 3 show that the result holds for any θ and *Q*. Thus, we have
$$Conjecture:\;V_{rev}\;is\;increasing\;in\;Q\;for\;l>r\;and\;decreasing\;in\;Q\;for\;l<r.$$

We remark that, in Figure 3, we take L=20mM, R=50mM, and $D_{1}=1.334\times 10^{-9}\;m^{2}/s$ and $D_{2}=2.032\times 10^{-9}\;m^{2}/s$ which are diffusion constants of, say, Na${}^{+}$ and Cl${}^{-}$, respectively (see the solid line), and $D_{1}=1.334\times 10^{-9}m^{2}/s$ and $D_{2}=0.792\times 10^{-9}m^{2}/s$, where $D_{2}$ is the diffusion constants of Ca${}^{2+}$ (see the dashed line).

#### 2.2.5. Dependence of $V_{rev}$ on $\rho =D_{2}/D_{1}$

Let us discuss the dependence of $V_{rev}$ on $\rho =D_{2}/D_{1}$ for effects of $D_{1}$ and $D_{2}$. It follows from Proposition 4.6 in [23] that
$$The\;reversal\;potential\;V_{rev}\;is\;increasing\;in\;\rho \;if\;l>r\;and\;is\;decreasing\;in\;\rho \;if\;l<r.$$

This feature reveals a fantastic aspect that is not intuitive immediately. Recall Equation (Equation 21). Given the boundary values and diffusion constants, the values one obtains for all terms in Equation (Equation 21) except $J_{k}$ are independent of $D_{k}$ [36]. The relation surely holds for the zero-current condition, i.e., $J_{1}=J_{2}$ with $V=V_{rev}$. Now, let us fix $D_{1}$ and increase $D_{2}$ (so ρ is increasing). Then, $|J_{2}|$ increases since all but $J_{2}/D_{2}$ in Equation (Equation 21) are independent of $D_{2}$. Consequently, to meet the zero-current condition, we need to increase $|J_{1}|$. Intuitively increasing $V_{rev}$ seems to lead to an increase in $|J_{1}|$. This intuition agrees with the result for l>r. However, for the case with l<r, the result is the exact opposite of the intuitive result. That is, for l<r, it says, as ρ increases, $V_{rev}$ decreases. This counterintuitive behavior could be explained by the fact that $c_{1}\left(x\right)$ depends on $V_{rev}$, and reducing $V_{rev}$ could still increase $|J_{1}|$. In fact, l<r will result in reducing $V_{rev}$, but $c_{1}\left(x\right)$ changes in a way that consequently increases $|J_{1}|$.

To illustrate the counterintuitive behavior, we provide a numerical result in Figure 4. We choose $C_{0}$, $\hat{L}$ and $D_{1}$ for Na${}^{+}$ as mentioned in Remark 1. Now, suppose that $D_{2}^{1}=0.792\times 10^{-9}m^{2}/s$, and consider the boundary concentrations L=20mM, R=50mM and Q=1M. In this case, $V_{rev}=-16.7657\;\mathrm{mV}$ and $J=-1.7632\times 10^{-17}\mathrm{mol}s^{-1}$. Now, if we increase $D_{2}^{1}$ to $D_{2}^{2}=2.032\times 10^{-9}m^{2}/s$, which is Cl${}^{-}$ diffusion constant, then $V_{rev}=-19.5527\;\mathrm{mV}$ and $J=-1.8788\times 10^{-17}\mathrm{mol}s^{-1}$. These values make sense now, based on the above discussion. Note that we just pictured the middle part of the channel in Figure 4 since the sides are almost identical. One should notice that it is hard to realize, from Figure 4, how L<R will result in reducing $V_{rev}$. The complicated behavior discussed above convinces us that detailed analytical studies, even for special cases, could be critical for the design and interpretation of numerical results.

### 2.3. A Comparison with Goldman–Hodgkin–Katz Equation for $V_{rev}$.

In this section, we first recall the GHK equation [14,15], which relates the reversal potential with the boundary concentrations and the permeabilities of the membrane to the ions. If the membrane is permeable to only one ion, then that ion’s Nernst potential is the reversal potential at which the electrical and chemical driving forces balance. The GHK equation is a generalization of the Nernst equation in which the membrane is permeable to more than just one ion. The derivation of GHK equation assumes that the electric field across the lipid membrane is constant (or, equivalently, the electric potential ϕ(x) is linear in *x* in the PNP model). Under the assumption, the I–V (current–voltage) relation is given by
$$I=V\sum_{k=1}^{n}z_{k}^{2}D_{k}\frac{r_{k}-l_{k}e^{z_{k}V}}{1-e^{z_{k}V}}.$$

For the case where n=2 and $z_{1}=-z_{2}=1$, the GHK equation for the reversal potential is
(23)$$\begin{array}{c} V_{rev}^{GHK}\left(\rho \right)=\ln\frac{r+\rho l}{l+\rho r}. \end{array}$$

The assumption that the electric potential ϕ(x) is linear is not correct when applied to channels in proteins. This is because proteins have specialized structure and spatial distributions of permanent charge (acid and base side chains) and polarization (polar and nonpolar side chains). *Experimental manipulations of the structure of channel proteins show that these properties control the biological function of the channel.* The GHK equation does not contain variables to describe any of these properties and so cannot account for the biological functions they control. A linear ϕ(x) is widely believed to make sense without channel structure presumably, in particular, where $Q_{0}=0$. However, this is not correct either. It follows from Formula (Equation 22) for $Q_{0}=0$ that the zeroth order in ε approximation of the reversal potential in this case is
(24)$$V_{rev}\left(0,\rho \right)=\frac{\rho -1}{\rho +1}\ln\frac{l}{r}.$$

Figure 5 compares $V_{rev}\left(0,\rho \right)$ in Formula (Equation 24) with $V_{rev}^{GHK}$ from the GHK-equation in Display (Equation 23). It can be seen from the left panel that, when *l* and *r* are not far away from each other (for example $L=C_{0}l=20\;\mathrm{mM}$, $R=C_{0}r=50\;\mathrm{mM}$), then the two curves have almost the same behavior. However, when we reduce *L* from 20mM to 1mM, then the right panel shows a significant difference between the two graphs.

In Figure 6, we arrange a simple numerical result for the case where Q≠0 to compare the graphs of $V_{rev}\left(Q,\rho \right)$, obtained from Formula (Equation 22), for various values of permanent charge Q. We consider L=20mM, R=50mM, and 0<ρ<5 for some values of Q, i.e., Q=0M,1M,10M.

### 2.4. Profiles of Relevant Physical Quantities

It follows that, for any given *Q*, once a solution (A,V) of Equations (Equation 15) and (Equation 16) is determined, all the other unknowns can be settled, and, hence, the approximation of the solution of boundary value problem can be obtained. We consider the dimensional form of quantities, and fix $\left(Q,L,R,D_{1},D_{2}\right)$ to numerically investigate the behavior of $C_{k}\left(X\right)$ and Φ(X) throughout the channel. Figure 7 and Figure 8 graph the cases with small permanent charge Q=0.1mM when L=20mM, R=50mM, $D_{1}=1.334\times 10^{-9}\;m^{2}/s,$ and $D_{2}=2.032\times 10^{-9}\;m^{2}/s$. In this case, we obtain $J=-1.2079\times 10^{-16}\;\mathrm{mol}s^{-1}$ and $V_{rev}=-4.4820\;\mathrm{mV}$.

Furthermore, Figure 9 and Figure 10 show graphs of concentrations, electrical potential, and electrochemical potentials versus *X*, where L=20mM, R=50mM, Q=2M, and diffusion constants are the same as the previous one. In this case, we obtain $J=-1.8789\times 10^{-17}\;\mathrm{mol}s^{-1}$ and $V_{rev}=-19.5527\;\mathrm{mV}$.

**Remark** **3.**
*We end this section with a few of the remarks on some important features captured in Figure 7, Figure 8, Figure 9 and Figure 10. It follows from the Nernst–Planck equation that $\mu_{k}^{\prime }\left(x\right)$ has the same sign as that of $\mu_{k}\left(1\right)-\mu_{k}\left(0\right)$ or the opposite sign as that of $J_{k}$; in particular, $\mu_{k}\left(x\right)$ is always monotonically increasing or decreasing. For the zero-current situation, the reversal potential depends on ALL other parameters; and so it seems that it would be hard to make general conclusions about $\mu_{k}\left(x\right)$, for example, about its monotonicity. This is not true. In fact, in Section 2.1, we have concluded that the sign of zero-current flux J is the same as that of L−R, and, hence, $\mu_{k}^{\prime }\left(x\right)$ has the opposite sign as that of L−R. For the case considered in this part, L=20mM<R=50mM, one has J<0, independent of the value of Q. Therefore, $\mu_{k}^{\prime }\left(x\right)>0$ for k=1,2, and, hence, $\mu_{k}\left(x\right)$’s are increasing for zero-current situation when L<R, independent of Q, as shown in Figure 8 for Q=0.1mM and in Figure 10 for Q=1mM. On the other hand, as one changes the value of Q, the profiles of concentrations $c_{k}\left(x\right)$’s and electrical potential ϕ(x) may vary from monotone to non-monotone, as shown between Figure 7 for Q=0.1mM and Figure 9 for Q=1M.*


## 3. Current–Voltage and Current-Permanent Charge Behaviors

Ionic movements across membranes lead to the generation of electrical currents. The current carried by ions can be examined through *current–voltage* relation or I–V curve. In such a case, voltage refers to the voltage across a membrane potential, and current is the flow of ions through channels in the membrane. Another important piece of data are *current-permanent charge* (I–Q) relation. Dependence of current on membrane potentials and permanent charge is investigated in this section for arbitrary values of diffusion constants.

To derive the I–V and I–Q relations, we rely on [33] where the authors showed that the set of nonlinear algebraic equations is equivalent to one nonlinear equation for *A*, the geometric mean of concentrations at x=a defined in Equation (Equation 17). All other quantities or variables such as fluxes, profiles of electric potential ϕ(x) and concentrations $c_{k}\left(x\right)$ can then be obtained in terms of *A*. It is crucial to realize that this is a specific result, not available for general cases. One can only imagine that the resulting simplification produces controllable and robust behavior that proved useful as evolution designed and refined protein channels. The reduction allowed by this composite variable can be postulated to be a “design principle” of channel construction, in technological (engineering) language, or an evolutionary adaptation, in biological language. In particular, the current *I* can be explicitly expressed in terms of boundary conditions, permanent charge, diffusion constants, and transmembrane potential in the special case that allows the determination of *A*. In what follows, we derive flux and current equations—when diffusion constants are involved as well—in terms of boundary concentrations, membrane potential, and permanent charge. The I–V, I–Q, J–V, and J–Q relations are investigated afterward in Section 3.2.

### 3.1. Reduced Flux and Current Equations

In this section, for simplicity, in addition to the assumptions at the beginning of the setup section (Section 1.2), we will also assume that h(x)=1, a=1/3 and b=2/3, in particular, α=1/3 and β=2/3 (see Display (Equation 19)). It was shown in [33] that the BVP (Equation 9) and (Equation 10) can be reduced to the algebraic equation
(25)$$\eta \ln\frac{S_{b}-\eta }{S_{a}-\eta }-N=0,$$
where B=l−A+r, $S_{a}$, $S_{b}$ and *N* are defined in Display (Equation 18), and
(26)$$\eta =Q_{0}-\frac{Q_{0}}{\ln\frac{Bl}{Ar}}\left(V+\ln\frac{l(S_{b}-Q_{0})}{r(S_{a}-Q_{0})}\right)+\frac{N}{\ln\frac{Bl}{Ar}}.$$

Once *A* is solved from Equation (Equation 25), we can obtain the flux densities and current equations as follows:(27)$$\begin{array}{cc} J_{k}:=J_{k}\left(V,l,r,D_{1},D_{2}\right)= & 3D_{k}\left(l-A\right)\left(1+{\left(-1\right)}^{k}\frac{\eta }{Q_{0}}\right),\;\mathrm{for}\;k=1,2,\\ I:=I(V,l,r,D_{1},D_{2})= & J_{1}-J_{2}=3\left(l-A\right)\left(D_{1}-D_{2}-\frac{\eta }{Q_{0}}\left(D_{1}+D_{2}\right)\right). \end{array}$$

For any given $\left(l,r,D_{1},D_{2},Q_{0},V\right)$, there exists a solution for the flux *J* and current *I*. The numerical results in the next section give us more information on “current–voltage” and “current-permanent charge” relations.

### 3.2. Current–Voltage and Current-Permanent Charge Relations

#### 3.2.1. Dependence of Current on Diffusion Constants

Now, we reveal a feature of the theoretical results that is not intuitive. Suppose that $\left(l,r,Q_{0}\right)$ is given (*V* is still free and is allowed to take any value!). It follows from Display (Equation 18) for the definition of *N* that there exists an *A* so that N=0. It consequently follows from Equation (Equation 26) that, if $V=\ln\frac{B(S_{a}-Q_{0})}{A(S_{b}-Q_{0})}$, then η=0. Therefore, from Display (Equation 27), $I=3\left(l-A\right)\left(D_{1}-D_{2}\right)$, which implies
$$For\;special\;values\;of\;parameters\;\left(l,r,V,Q_{0}\right),\;the\;sign\;of\;I\;is\;determined\;by\;the\;sign\;of\;D_{1}-D_{2}.$$

#### 3.2.2. I–V Curves and I–Q Curves

Figure 11 is a numerical simulation from Equations (Equation 25) and (Equation 26) of the I–V curves for several values of Q with $D_{1}=1.334\times 10^{-9}\;m^{2}/s$ and $D_{2}=2.032\times 10^{-9}\;m^{2}/s$. One may suspect, based on the numerical observations, that the value of current I, obtained from Display (Equation 27), is unique for any V, and is monotonically increasing in V. However, this may not be correct, in general. This is important since the opening and closing properties of channels might be thought to arise from non-unique solutions [16,17].

Now, for I–Q relations, our numerical experiments shows that:I–Qcurvesarenotmonotonicingeneral.

Recall that Equation (Equation 21), in dimensional form, gives
$$J_{k}\int_{0}^{\hat{L}}\frac{k_{B}T}{D_{k}A\left(X\right)C_{k}\left(X\right)}dX=\mu_{k}\left(0\right)-\mu_{k}\left(\hat{L}\right),\;k=1,2.$$

The sign of $J_{k}$ is determined by the boundary conditions, independent of the permanent charge. Nevertheless, as expected and seen in Figure 12, the magnitudes of $J_{k}$’s, and, consequently, the sign and the size of the current I do depend on $Q=2Q_{0}C_{0}$ in general. (Here, Q would be the nonzero value of the permanent charge in dimensional form.) Treating V as a parameter, the current I is a function of Q. The numerical observations in Figure 12 indicate that,
(i)there exists some $V^{*}$ such that, for $V>V^{*}$, I(Q) has a unique maximum;(ii)there exists some $V_{*}$ such that, for $V<V_{*}$, I(Q) has a unique minimum.

In particular, I–Q curves are not monotonic in general.

In addition, we claim based on numerical observations (not proven though) that there exists $\hat{V}\left(D_{1},D_{2}\right)=\min\{V^{*},|V_{*}\left|\right\}$, such that
(i)for any given V where $|V|>\hat{V}$, the corresponding current I is non-monotonic in Q, but(ii)for any V where $|V|<\hat{V}$, the corresponding current I is monotonic in Q.

In particular, it can be seen in Section 3.3 that current is monotonic in Q for V=0. In the end, we would like to mention that the diffusion constants affect the values $V^{*}$ and $V_{*}$ above.

### 3.3. Zero-Voltage Current

The different permeability of the membrane determines the zero membrane potential (voltage) to different types of ions, as well as the concentrations of the ions, the permanent charge, and the shape of the channel. Denote current I(V;Q), and the fluxes $J_{k}\left(V;Q\right)$, for k=1,2, to include the dependence on *Q* too. We call I(0;Q) the *zero-potential current* and $J_{k}\left(0;Q\right)$ the *zero-potential fluxes*, respectively, when V=0. For any given value of membrane potential *V*, approximation formulas for the current I(V;Q), for small and large values of permanent charge *Q*, are provided in [34,38], respectively.

It follows from [34] that, for *small* values of *Q*, applying V=0, zero-potential current $I^{s}\left(0;Q\right)$, and zero-potential fluxes $J_{k}^{s}\left(0;Q\right)$ (in dimensionless forms as mentioned in Remark 2) are
(28)$$\begin{array}{ll} I^{s}\left(0;Q\right)= & \left(l-r\right)\left(D_{1}-D_{2}\right)-\frac{3}{2}\frac{\left(D_{1}+D_{2}\right){\left(l-r\right)}^{2}}{\left(2l+r\right)\left(l+2r\right)\ln\frac{l}{r}}Q+O\left(Q^{2}\right),\\ J_{k}^{s}\left(0;Q\right)= & \left(l-r\right)D_{k}+{\left(-1\right)}^{k}\frac{3{\left(l-r\right)}^{2}D_{k}}{2\left(2l+r\right)\left(l+2r\right)\ln\frac{l}{r}}Q+O\left(Q^{2}\right),\;k=1,2. \end{array}$$

For *large* positive values of $Q=2Q_{0}$, with $\nu =1/Q_{0}$ (where ν is small), it follows from [38] that zero-potential current $I^{l}\left(\nu \right)=I^{l}\left(0;Q\right)$ and zero-potential fluxes $J_{k}^{l}\left(\nu \right)=J_{k}^{l}\left(0;Q\right)$ are
(29)$$\begin{array}{ll} I^{l}\left(\nu \right)= & -6D_{2}\sqrt{lr}\frac{\left(\sqrt{l}-\sqrt{r}\right)}{\sqrt{l}+\sqrt{r}}+\frac{3}{2}D_{1}{\left(\frac{l+r}{\sqrt{l}+\sqrt{r}}\right)}^{2}\left(l-r\right)\nu \\ \; & +\frac{3}{2}D_{2}\frac{l+r}{{\left(\sqrt{l}+\sqrt{r}\right)}^{2}}f\left(l,r\right)\left(l-r\right)\nu +O\left(\nu^{2}\right),\\ J_{1}^{l}\left(\nu \right)= & \frac{3}{2}D_{1}{\left(\frac{l+r}{\sqrt{l}+\sqrt{r}}\right)}^{2}\left(l-r\right)\nu ,\\ J_{2}^{l}\left(\nu \right)= & 6D_{2}\sqrt{lr}\frac{\sqrt{l}-\sqrt{r}}{\sqrt{l}+\sqrt{r}}-\frac{3}{2}D_{2}\frac{l+r}{{\left(\sqrt{l}+\sqrt{r}\right)}^{2}}f\left(l,r\right)\left(l-r\right)\nu +O\left(\nu^{2}\right), \end{array}$$
where
(30)$$\begin{array}{cc} f(l,r)= & \frac{2lr}{{\left(\sqrt{l}+\sqrt{r}\right)}^{2}}+l+r-\frac{1}{2}\frac{\ln l-\ln r}{l-r}{\left(l+r\right)}^{2}. \end{array}$$

It can be readily seen from Equations (Equation 28) that, for small values of *Q*, the zero-potential current $I^{s}\left(0;Q\right)$ is increasing in *Q* when l<r and is decreasing in *Q* if l>r.

However, for large values of the permanent charge *Q*, the zero-potential current $I^{l}\left(0;Q\right)$ depends on *Q* in a much richer way. To state the results, we need the following lemma.

**Lemma** **1.**
*There are $t_{1}$ and $t_{2}$ with $0<t_{1}<1<t_{2}$ so that f(l,r)>0 for $l/r\in (t_{1},t_{2})$ and f(l,r)<0 for $l/r\notin [t_{1},t_{2}]$.*


Note that
$$\frac{d}{d\nu }I^{l}\left(0\right)=\frac{3D_{2}}{2}{\left(\frac{l+r}{\sqrt{l}+\sqrt{r}}\right)}^{2}\left(\frac{D_{1}}{D_{2}}+\frac{f(l,r)}{l+r}\right)\left(l-r\right).$$

It follows from Equations (Equation 29) and Lemma 1 that, for large values of *Q* (small values of ν),
(i)if $l/r\in [t_{1},t_{2}]$ (so that f(l,r)≥0), then, for arbitrary $D_{1}$ and $D_{2}$, the zero-potential current $I^{l}\left(\nu \right)$ is decreasing in ν (increasing in *Q*) when $l/r\in [t_{1},1)$, and is increasing in ν (decreasing in *Q*) when $l/r\in (1,t_{2}]$;(ii)if $l/r\notin [t_{1},t_{2}]$ (so that f(l,r)<0), then,
(a)for $\frac{D_{1}}{D_{2}}+\frac{f(l,r)}{l+r}>0$, the zero-potential current $I^{l}\left(\nu \right)$ is decreasing in ν (increasing in *Q*) for l<r, and is increasing in ν (decreasing in *Q*) for l>r;(b)for $\frac{D_{1}}{D_{2}}+\frac{f(l,r)}{l+r}<0$, the zero-potential current $I^{l}\left(\nu \right)$ is increasing in ν (decreasing in *Q*) for l<r, and is decreasing in ν (increasing in *Q*) for l>r.

Figure 13 illustrates some of the above conclusions. In addition, it suggests that the monotonicity of I(0) holds for all values of permanent charge, not only for small or large values. We emphasize that the monotonicity of current *I* with respect to permanent charge *Q* is just true for zero membrane potential, i.e., V=0. Indeed, one should recall from Section 3.2 that, when V≠0, then the current *I* is not monotonic in *Q*.

## 4. Conclusions

In this paper, we first recall the analytical results in [23] for arbitrary diffusion constants. To investigate the reversal potential problem for which the current is zero, we do numerical investigations based on the analytical results in [23], where many cases are studied analytically. We derive several remarkable properties of biological significance, from the analysis of these governing equations that hardly seem intuitive.

Biophysicists are also interested in the relation of current–voltage (I–V), and current-permanent charge (I–Q), as well as reversal potential problems. To do that, we first recall the analytical results in [33], for arbitrary diffusion constants, to drive the flux densities and current equations explicitly. One way to characterize channels is the current at zero electric potential, that is, when V=0, which has practical advantages. Since it is usually easier to measure a large current than a vanishing one, we analyzed this case as well. Furthermore, we briefly study the special cases of small and large permanent charge for zero voltage case, based on the analytical results of [34,38], respectively. To bridge between small and large values of permanent charges, we numerically study I–V and I–Q relations for this case as well.

## Figures and Tables

**Figure 1 entropy-22-00325-f001:**
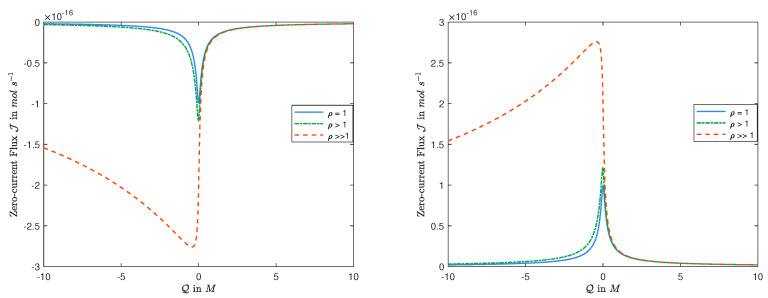
The function J=J(Q) for various values of $\rho =D_{2}/D_{1}$: The left panel for L=2mM and R=5mM; the right panel for L=5mM and R=2mM.

**Figure 2 entropy-22-00325-f002:**
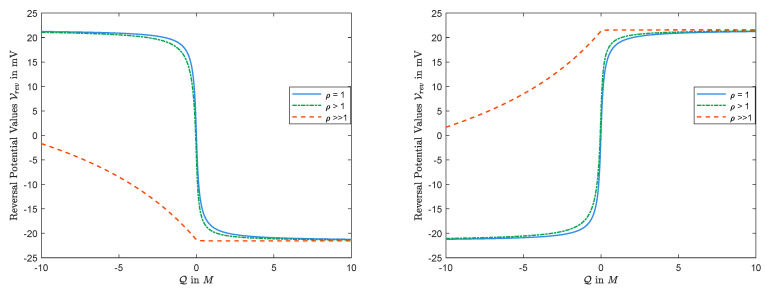
The function $V=V_{rev}\left(Q\right)$: The left panel for L=2mM and R=5mM; the right panel for L=5mM and R=2mM.

**Figure 3 entropy-22-00325-f003:**
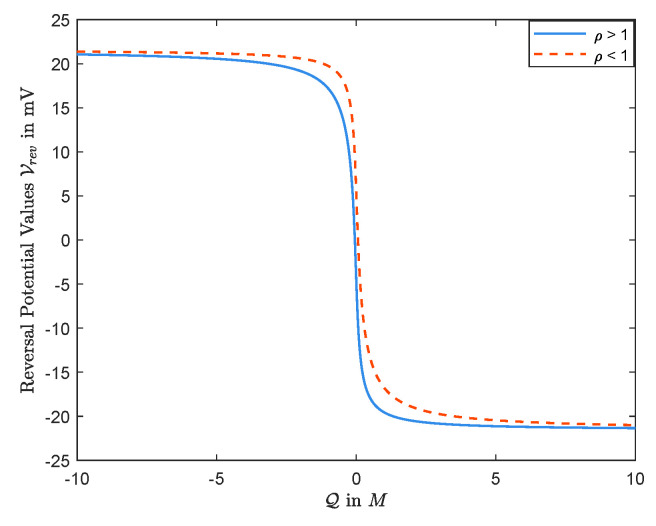
$V=V_{rev}\left(Q\right)$ decreases when L<R, independent of values of diffusion constants.

**Figure 4 entropy-22-00325-f004:**
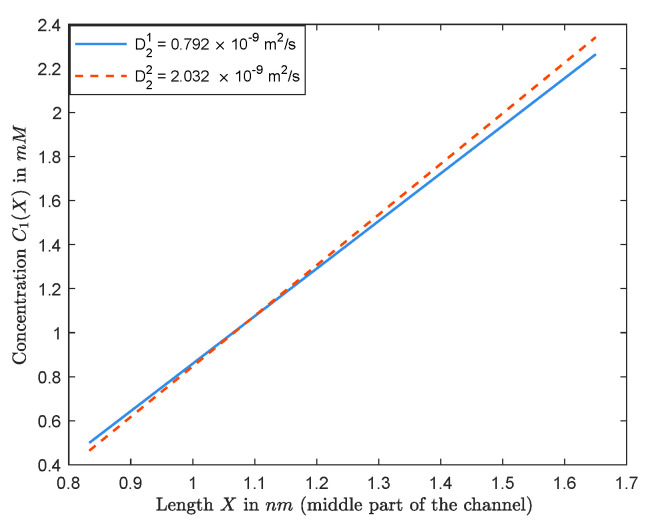
Graphs of $C_{1}\left(X\right)$ when $D_{1}$ is fixed, but we increase $D_{2}$.

**Figure 5 entropy-22-00325-f005:**
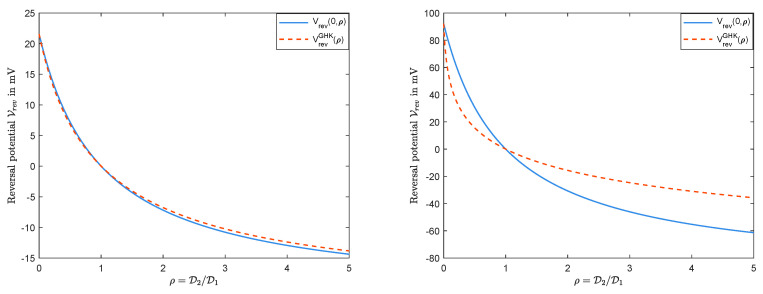
$V_{rev}\left(Q=0,\rho \right)$ vs. $V_{rev}^{GHK}\left(\rho \right)$: The left panel for L=20mM and R=50mM; the right panel for L=1mM and R=50mM.

**Figure 6 entropy-22-00325-f006:**
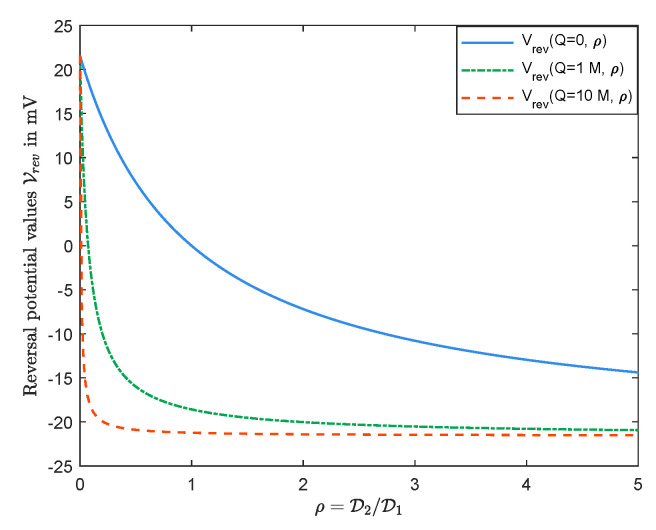
$V_{rev}\left(Q,\rho \right)$ with various values of permanent charges.

**Figure 7 entropy-22-00325-f007:**
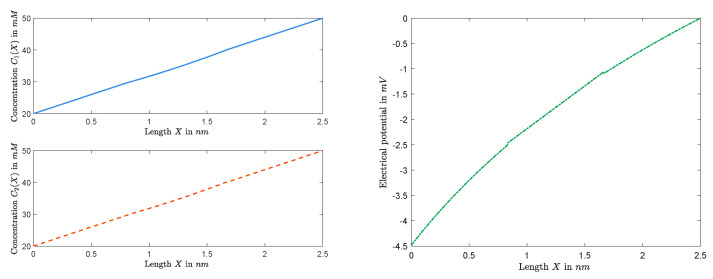
The functions $C_{k}\left(X\right)$ (**left**) and Φ(X) (**right**) with Q=0.1mM.

**Figure 8 entropy-22-00325-f008:**
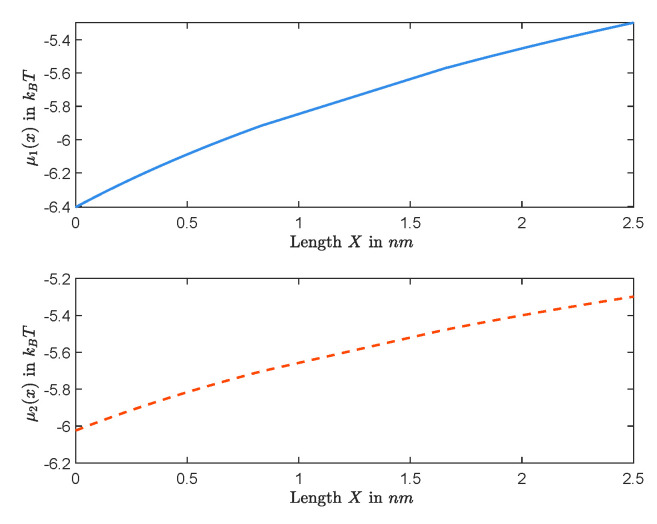
The functions $\mu_{1}\left(X\right)$ and $\mu_{2}\left(X\right)$ are increasing for Q=0.1mM.

**Figure 9 entropy-22-00325-f009:**
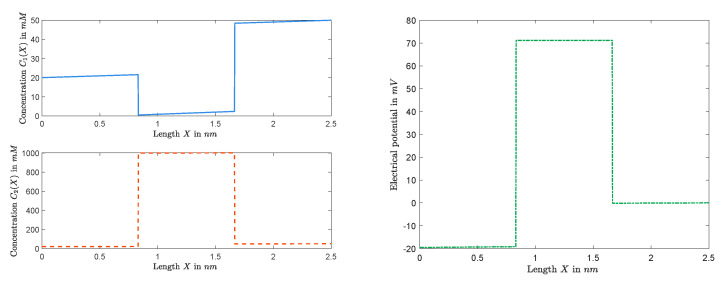
The functions $C_{1}\left(X\right)$ and $C_{2}\left(X\right)$ (**left**) and the function Φ(X) (**right**) for Q=1M.

**Figure 10 entropy-22-00325-f010:**
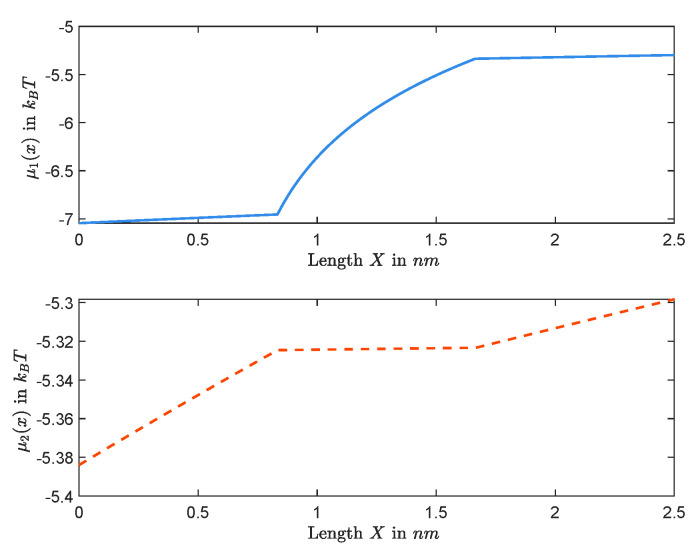
The functions $\mu_{1}\left(X\right)$ and $\mu_{2}\left(X\right)$ are increasing for Q=1M.

**Figure 11 entropy-22-00325-f011:**
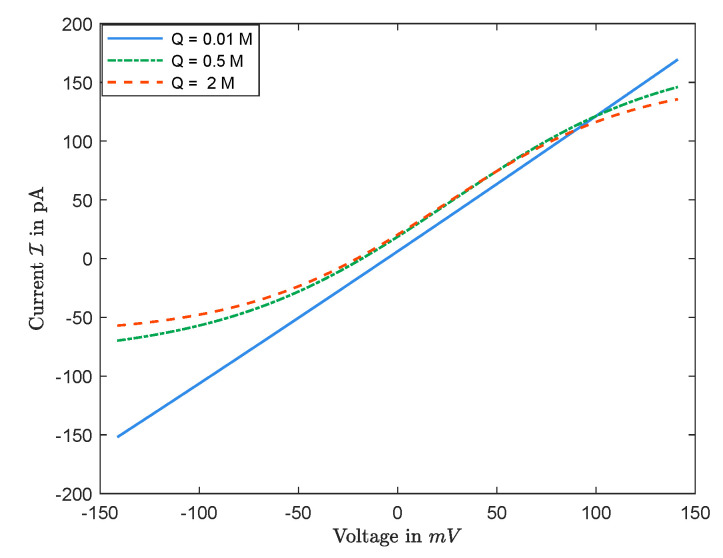
The function I=I(V) for L=20mM and R=50mM.

**Figure 12 entropy-22-00325-f012:**
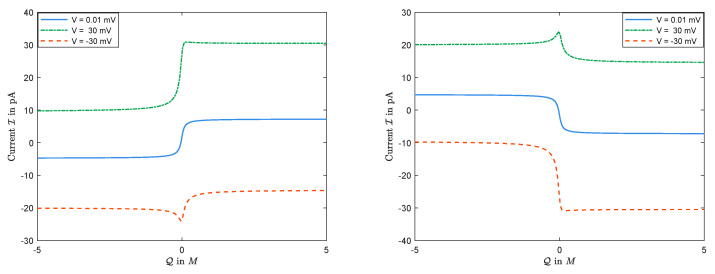
The function I=I(Q) with $D_{1}<D_{2}$: The left panel for L=20mM and R=30mM; the right panel for L=30mM and R=20mM.

**Figure 13 entropy-22-00325-f013:**
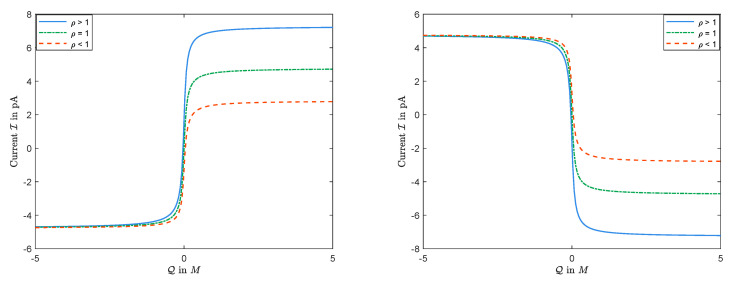
The function I=I(Q) for V=0: The left panel for L=20mM and R=30mM; the right panel for L=30mM and R=20mM.

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
