# Peer review of "Effects of Diffusion Coefficients and Permanent Charge on Reversal Potentials in Ionic Channels"

_entropy, 2020, doi:10.3390/e22030325_

Round 1

Reviewer 1 Report

The manuscript reports important and essential developments in the theoretical description of ion transport in narrow pores. The authors present a comprehensive analysis of the influence of physiological parameters on the transport and made several predictions on how the change of the pore and other parameters will affect the experimentally measured quantities. The manuscript is well written, and I recommend the manuscript to be published in the journal Entropy. However, I think that the manuscript requires a minor revision before publishing. Below I provide few points to consider.

P1. The authors do not mention that their approaches can be applied to artificial nanopores. What is a barrier to this?

P2. References are used in arbitrary order in the text, so it is not possible to verify that all references have been used. Is it possible to correct the order?

P3. The authors repeat several times that many results in the paper are not intuitive. Is it possible to provide one or two examples earlier in the text, for example in the abstract or the introduction. I think that the highlights (page 6, lines 152-168) should be in the introduction, otherwise they look as a remark.

P4. On page 2, lines 41-42, the authors mentioned “individual fluxes”. Can the authors provide examples of any measurements of the fluxes in ion channels? In solid-state physics, the measurement of the flux of individual carrier (typically electron) is a hot topic, which often related to characterizing fluctuational statistics of the flux, so-called full counting statistics. Do the authors know any similar developments in connections to ion channels? Can the authors provide corresponding references?

P5. In the introduction, the authors mention that they solve a system of nonlinear equations, does the used Matlab method guarantee the uniqueness of the solution, or how do the authors check the uniqueness of the solution? Did the authors observe multiple solutions?

P6. In Eq. (1) on page 3 and other following equations, the authors use subscript index “s” in Poisson equation and index “k” in Nernst-Plank equation. Index “k” has been explained, and it is a useful reference point for understanding text. Is it possible to use “k” instead “s” in Poisson equation as well? Then the use of the same index would link the equations better.

P7. Could the authors clarify the application of the singular limit? In the text, the authors use \epsilon as a small parameter, but Eq. (9) on page 4 has a different parameter: \frac{\epsilon^2}{h(x)}, that is function h(x) included. Does the use of \epsilon only mean that first equation in (9) was multiplied by h(x) before the application of the singular limit?

P8. Do the authors compare the solution with a finite value of \epsilon, that can be obtained by solving BVP (9) and (10) on page 4, with the solution obtained via the singular limit? What is the difference, what is the role of the finite value of \epsilon?

P9. Page 7, line after line 193, “cPNP” should be replaced by “PNP”.

P.10. I think that the authors should change the numeration of the equations: currently, the same numbers are used for equations from different sections. As a result, it’s not possible to infer which number is referred to which expression. The authors could include the section number in the numeration: (1.X) for first section and (2.X) for second and (3.X) for third.

P.11. In several figures “Q in M” is used as x-label. I was not able to find out what is the meaning of this notation.

P.12. The authors often use “variable”, is it possible to use terminology that helps to see what this variable is? For example, instead “variable A” use “geometric mean of concentrations” A. Another example, instead “function C_k(x)” use “concentration C_k(x)”. It would help to see the results in the biological context.

P.13. Page 11, line 286. I think that “the whole truth” cannot be derived from an approximate consideration. Could the authors rephrase the phrase?

P.14. Page 12, line 296. The authors use solution (A,V), do they mean “concentration as a function of voltage” or “a point (A,V)”?

P.15. Page 13. Line 317. The authors state that the profiles may vary from 317 monotone to non-monotone, as shown between Figure 7. I can’t see a non-monotonic curve in Fig. 7. Could the authors clarify their statement.

P.16. Page 14. Lines 318-331. The text is a copy of some previous part of the text. I think it should be removed.

P.17. Page 16. Lines 362-368. The authors mention multiple solutions. First, I do not see any illustrations of such a solution. Second, does it mean that some solutions are stable and other are unstable?

P.18. I think that the curve is “monotonic”, not “monotone”.

P.19. Page 17, lines after Fig. 14 up to 375. I can’t see any links of this part of the text to the previous paragraph.

P.20. Lines 389-390. There is no enough space between the lines.

P.21. Is it correct that the presented results can be mapped against experimental results? If so, are such experimental results are available or the need to be done? Could the authors add a brief discussion of these questions in Conclusion?

Reviewer 2 Report

Mofidi et al. used the Nerntst-Plank-Poisson equations of ionic transport to derived a relationship between the reversal potential and the diffusion coefficient of ions. Goldman-Hodgkin-Katz (GHK) equation is normally used in ion channels to calculate the reversal potential from the permeability of ions. The authors here claim that the constant field approximation used in the GHK equation is incorrect and they present an alternative equation for reversal potential in equation 7. However, the equation derived here is much more complex than the GHK equation and unlikely to be practical. The bigger problem though is the validity of the results in this paper. The results are not tested against any experimental data or numerical simulations. Furthermore, this manuscript is a follow up of a previous preprint by two of the authors (arXiv:1909.01192] and the results in arXiv:1909.01192 is the starting point for the derivation on this work. Since the previous manuscript (arXiv:1909.01192) hasn’t been reviewed and published, I cannot accept the result taken from there as correct. My concern is that if the results in the earlier preprint are wrong, then the results in this paper are wrong as well. I’m not saying that those results are wrong, but that they need to be verified first. However, the earlier preprint is too long and mathematically involved for me to read through and verify its correctness in a reasonable time. I think the authors should wait until the earlier preprint is published before they submit this paper for publication.

Reviewer 3 Report

Revisiting the biophysical foundations and the accuracy of the GHK equation regarding the description of ionic currents and fluxes through transmembrane channels is an important but often overloooked topic. A rigourous treatment of this problem as performed by the authors is to be commended.

  However, in my opinion, the major issue with the manuscript is whether the PNP equations are still valid within ionic channels.  Indeed, the radius of the channel is typically so small that it can block the passage of some ions altogether.  The hydration shell can also be modified during the passage of the ion through the channel.  A model considering ions as discrete quantities would probably be preferable.  The hypothesis that the charge density is piecewise constant also seems a  bit dubious to me. In other words, the channel model used here may be a misleading oversimplification. At the very least, the authors should discuss 1) why is treating ions as a continuous quantity (ie  using concentration and PNP equations) is appropriate within a channel? 2) How is the hypothesis of piecewise constant charge density justified?

This manuscript seems to rely heavily on previous work by the authors.  While this is perfectly okay, in order to facilitate the task of the unfamiliar reader, the authors should provide more details on the relevant parts of their previous works instead of only giving references.

If the authors submit a revised version of the manuscript, I encourage them to pay attention to the quality of English since I found a few typos without particularily looking for them.

I wish the author the best of luck with their work

Reviewer 4 Report

This work conducts a rigorous mathematical analysis on properties of ionic flows through membrane channel. Based on the authors' previous analytical results using geometric singular perturbation analysis of the classical Poisson-Nernst-Planck models for symmetric 1:1 electrolyte, this study summarizes and extends the results in details on the dependences of reversal potential, zero-potential flux/current on permanent charges, voltage, ionic diffusion constants and concentrations. Though, the setup of the analyzed system is a mathematical simplification and idealization, the conditions are biological relevant situations and the derive properties are of biological significance and remarkable. At the same time, these results are truly not intuitive, and would bring in-depth insights in channel studies.

Here are two issues in the simplified model. One is that the diffusion coefficient of an ion is usually not constant in channel study. It may even happen that the ratio between diffusion coefficients of two different ion species can even qualitatively change from bulk region to pore region (e.g., possible due to change of ionic solvation shell). Another issue is that in many three dimensional cases the permanent charges are surface charges. Is any qualitative consequences to change the setup from 2D surface charges to 3D volume charges?

Round 2

Reviewer 1 Report

I thank the authors for their careful revision of the manuscript and addressing each single comment.

Artificial Nanopore. I think that  the following paper is relevant:

M. Zwolak, J. Lagerqvist, and M. Di Ventra, “Quantized ionic conductance in nanopores,” Phys. Rev. Lett., vol. 103, p.
128102, 2009.

Full Counting Statistics. In fact, this is a continuation of old topic of "shot noise" in electronic devices, but with respect to a single carrier: electron. I think that this paper should give an idea of this topic:

TheBlanter, Y.M.; Büttiker, M. Shot noise in mesoscopic conductors. Phys. Rep. 2000, 336, 1–166.

Note, that I am not aware any analysis of statistics of passing ions in pore.

Reviewer 2 Report

While the authors made a few changes, (mainly in response to comments from other referees) my overall concern remains the same. It is not possible to understand the paper without understanding their unpublished paper. I even have to look up their previous paper to understand the notations. For example, the authors don't define the terms \alpha and \beta used in equation (4) on page 8. And even where they are defined, the authors put no effort to explain what they mean. This would be fine for purely mathematical journal but not for physical science. In some instances, they use the same symbols for two different quantities, for example, 'A' is used for the area on pages 4-5, and mean concentration on page 8. All, these factors make the paper incomprehensible. 

The author should also cite the classic paper on the effect of diffuion on ion transport through tiny pores by Lauger: Biochimica et Biophysica Acta, 455, 493-509 (1976). He does a similar analysis but in a much intuitive way.

Author Response

Please the attachment.

Reviewer 3 Report

I thank the authors for their response.  I still would like them to include a discussion about the limitations of 1) The use of the PNP formalism in the context of single channel modelling.  2) The description of channels as cylinders with uniform width and uniform charge distribution.

With respect to point 1), the paper : Maffeo C, Bhattacharya S, Yoo J, Wells D, Aksimentiev A. Modeling and simulation of ion channels. Chem Rev. 2012;112(12):6250–6284. doi:10.1021/cr3002609  provides a good discussion of the strengths and limitation of this approach.  I very well understand that one should not model indivudal ions with the PNP formalism.  That's exactly a limit of the approach. The use of 'concentrations'  in itself is a simplification.

Hyon, Y., Eisenberg, B. & Liu, C. (2010), ‘A mathematical model of the hard sphere repulsion in ionic solutions’, Communications in Mathematical Sciences 9, 459–475. also discusses the impact of the size of the ions in the modelling of channels.

With respect to point 2) a large body of experimental work has investigated the geometry of channels (and how the differ from simple cylinders) and the charge distribution within these channels.  For example,  potassium channels possess a selectivity filter which is important (without it they would be permeable to sodium).

I believe the authors should discuss these limitations of their work (which is nice from a mathematical point of view) in order not to be misleading with respect to the impact of their work.

Best of luck

Reviewer 4 Report

Accept

Author Response

Thanks!

Round 3

Reviewer 2 Report

I don't have any major comment beside the ones that I've already mentioned in the previous reviews. The authors are not willing/able to address them at this time. So, I leave up to the editor to make a decision.

Line 44: What do the authors mean by "dangerous to measure [individual flux]? Do you mean literally or figuratively? 

Reviewer 3 Report

The authors made major modifications and improvements to the manuscript.

The mathematical treatment is of high quality.

This is certainly quality work deserving publication.